# *Citrullus mucosospermus* Extract Reduces Weight Gain in Mice Fed a High-Fat Diet

**DOI:** 10.3390/nu16132171

**Published:** 2024-07-08

**Authors:** He Mi Kang, Sun Young Park, Ji Eun Kim, Ki Won Lee, Dae Youn Hwang, Young-Whan Choi

**Affiliations:** 1Department of Horticultural Bioscience/Life and Industry Convergence Research Institute, College of Natural Resources and Life Science, Pusan National University, Miryang 50463, Republic of Korea; mimi2965@naver.com; 2Institute of Nano-Bio Convergence, Pusan National University, Busan 46241, Republic of Korea; sundeng99@pusan.ac.kr; 3Department of Biomaterials Science (BK21 FOUR Program)/Life and Industry Convergence Research Institute, College of Natural Resources and Life Science, Pusan National University, Miryang 50463, Republic of Korea; prettyjiunx@naver.com (J.E.K.); dyhwang@pusan.ac.kr (D.Y.H.); 4Natural Products Convergence R&D Division, Kwangdong Pharma. Co., Ltd., Seoul 08381, Republic of Korea; wish2k@nate.com

**Keywords:** *Citrullus mucosospermus*, anti-obesity effects, anti-adipogenesis

## Abstract

This study aimed to investigate the therapeutic potential of *Citrullus mucosospermus* extract (CME) in counteracting adipogenesis and its associated metabolic disturbances in murine models. In vitro experiments utilizing 3T3-L1 preadipocytes revealed that CME potently inhibited adipocyte differentiation, as evidenced by a dose-dependent reduction in lipid droplet formation. Remarkably, CME also attenuated glucose uptake and intracellular triglyceride accumulation in fully differentiated adipocytes, suggesting its ability to modulate metabolic pathways in mature adipose cells. Translating these findings to an in vivo setting, we evaluated the effects of CME in C57BL/6N mice fed a high-fat diet (HFD) for 10 weeks. CME administration, concomitantly with the HFD, resulted in a significant attenuation of body weight gain compared to the HFD control group. Furthermore, CME treatment led to substantial reductions in liver weight, total fat mass, and deposits of visceral and retroperitoneal adipose tissue, underscoring its targeted impact on adipose expansion. Histological analyses revealed the remarkable effects of CME on hepatic steatosis. While the HFD group exhibited severe lipid accumulation within liver lobules, CME dose-dependently mitigated this pathology, with the highest dose virtually abolishing hepatic fat deposition. An examination of adipose tissue revealed a progressive reduction in adipocyte hypertrophy upon CME treatment, culminating in a near-normalization of adipocyte morphology at the highest dose. Notably, CME exhibited potent anti-inflammatory properties, significantly attenuating the upregulation of pro-inflammatory cytokines’ mRNA levels (TNF-α, IL-1β and IL-6) in the livers of HFD-fed mice. This suggests a potential mechanism through which CME may exert protective effects against inflammation associated with obesity and fatty liver disease.

## 1. Introduction

Obesity is a significant, dangerous factor for several diseases, including type 2 diabetes, cardiovascular disease, hypertension, Alzheimer’s disease, asthma, metabolic syndrome, fatty liver disease, gallbladder disease, osteoarthritis, obstructive sleep apnea, various cancers, hypercholesterolemia, musculoskeletal disorders, and nonalcoholic fatty liver disease (NAFLD) [1,2]. The link between obesity and these conditions can be attributed to several factors such as inflammation, metabolic and mitochondrial dysfunction, hyperinsulinemia, and abnormal lipid and glucose metabolism. Dysfunctional adipose tissues are central to the development of these diseases [3,4]. They release cytokines and bioactive substances that drive inflammation, insulin resistance, and endothelial dysfunction, all contributing to obesity-related diseases. Additional factors include chronic inflammation, abnormal angiogenesis, interactions among inflammatory cytokines, endocrine hormones, adipokines, genetic predispositions, and environmental influences [5,6,7,8]. Prolonged consumption of a high-calorie, high-fat diet has been found to cause excessive fat accumulation in the body’s white adipose tissue [9,10]. This finding suggests a profound correlation between excessive fat production and chronic diseases associated with obesity. White adipose tissue is composed of adipocytes, and the number of adipocytes increases as preadipocytes differentiate into mature fat cells [11,12]. To observe the processes of adipogenesis and adipocyte differentiation under laboratory conditions, the 3T3-L1 murine adipocyte cell line is widely utilized by researchers [13].

*Citrullus lanatus*, commonly known as the Sweet Dessert Watermelon, is a prominent gourd plant that originates from tropical and subtropical locales, establishing itself as a pivotal vegetable crop with widespread global consumption [14,15]. Renowned for its nutritional bounty, encompassing water, carotenoids, organic acids, vitamins, carbohydrates, fats, crude fiber, amino acids, nucleotides, citrulline, arginine, lycopene, and cucurbitacins, this botanical marvel resonates with therapeutic and pharmacological significance [16,17]. Within the Cucurbitaceae family, the *Citrullus* genus contains diverse species including *C. lanatus* and *Citrullus mucosospermus*, colloquially known as Egusi Watermelon. Indigenous to the African continent, this variant is predominantly cultivated for its delectable seeds, which eschew the hard seed coat and render them palatable in their raw form [18,19]. In contrast, the white fruit pulp manifests notable bitterness, relegating it to direct consumption. The recent scientific exploration of *C. mucosospermus* has revealed its multifaceted potential, showing attributes such as antioxidant and anti-inflammatory effects, anticancer properties, antimicrobial activities, blood sugar regulation, cholesterol control, and weight loss implications [20,21,22]. This botanical gem holds a venerable position in the tapestry of traditional African cuisine, with ongoing research revealing its diverse benefits [23]. Despite both being *Citrullus*, *C. lanatus* and *C. mucosospermus* have distinctive characteristics, including seed edibility, fruit pulp flavor, nutritional composition, and culinary applications [24,25]. *Citrullus lanatus* is consumed globally due to its succulent and sweet fruit, in its raw state or in juices, salads, and desserts [26,27]. In contrast, *C. mucosospermus* seeds are used primarily in traditional African culinary practices due to their bitterness. The ongoing trajectory of present research positions *C. mucosospermus* as an emerging focal point, revealing its potential efficacy across a myriad of domains [15,17,19].

The purpose of this study was to determine the effects of *Citrullus mucosospermus* extract (CME) on lipogenesis, liver fat accumulation, and inflammatory response in an HFD-induced obesity mouse model.

## 2. Materials and Methods

### 2.1. Reagents and Chemicals

For our research, we procured a variety of reagents and chemicals from several suppliers. Insulin, dexamethasone, rosiglitazone, HEPES, Oil Red O, NaHCO3, phenylmethylsulfonyl fluoride, and sodium deoxycholate were obtained from Sigma-Aldrich (St. Louis, MO, USA). GENEray (Shanghai, China) supplied sodium dodecyl sulfate (SDS), isopropanol, and Tween 20. paraformaldehyde was sourced from TCI (Tokyo, Japan). The protein assay reagents, Western blotting loading buffer, and protein markers were purchased from Bio-Rad (Hercules, CA, USA). Additionally, glucose detection kits and triglyceride (TG) were acquired from Asan Pharmaceutical (Seoul, Republic of Korea).

### 2.2. Extraction of Citrullus mucosospermus

*Citrullus mucosospermus* plants were grown at the Pusan National University farm, with planting occurring in mid-April. Once the fruits reached full ripeness, they were harvested and subjected to freeze drying using an Ilshin freeze dryer (Dongducheon, Republic of Korea). The dried fruits were then ground into a fine powder. This powder was sifted through a 30-mesh sieve to ensure uniform particle size and stored at −20 °C until extraction. For the extraction process, 10 g of the powdered sample was mixed with distilled water at a ratio of 1:10 (*w*/*v*). This mixture was treated with an ultrasonicator (JAC Ultrasonic, Seoul, Republic of Korea) for one hour at an ambient temperature. The resulting extract was then filtered through Whatman No. 2 filter paper (Whatman International Ltd., Maidstone, UK). Following filtration, the solvents were evaporated using a rotary evaporator (Heidolph, Heidolph Korea Ltd., Seoul, Republic of Korea), yielding 4.46 g of the water-based extract. The *Citrullus mucosospermus* extract (CME) was subsequently stored in a glass container at −20 °C until it was required for further experimentation.

### 2.3. HPLC Analysis

The quantities of cucurbitacin E and cucurbitacin *E*-2-*O*-glucoside in the *Citrullus mucosospermus* extract (CME) were measured using an Agilent 1100 high-performance liquid chromatography (HPLC) system (Santa Clara, CA, USA). This system featured an autoinjector and a column temperature controller. The separation was carried out on a reversed-phase Luna C18 column (150 × 4.6 mm, 5 μm particle size, Phenomenex, Torrance, CA, USA). For the mobile phase, 0.01% formic acid in water (solvent A) and acetonitrile (solvent B) were used. The gradient elution was programmed as follows: starting with 40% solvent B at 0 min, maintaining 40% B from 0 to 30 min, and then increasing to 100% B from 30 to 40 min. A 20 μL aliquot of CME was injected into the system, with the flow rate set at 0.5 mL/min. The column temperature was consistently kept at 30 °C during the analysis.

### 2.4. Adipocyte Culture and Differentiation

Adipocyte culture and differentiation were conducted using the 3T3-L1 cell line obtained from the Korea Cell Line Bank (KCLB, Seoul, Republic of Korea). Cells were maintained in Dulbecco’s Modified Eagle’s Medium (DMEM) supplemented with 10% fetal bovine serum at 37 °C in a 5% CO_2_ atmosphere. Upon reaching 80% confluence, cells were enzymatically dissociated and seeded into various culture plates. They were then allowed to adhere and stabilize over a two-day period. The experimental groups included untreated 3T3-L1 cells as the control group and cells treated with a differentiation medium (DMI) as the model group. The DMI consisted of 10 μg/mL insulin, 1 μM dexamethasone, and 10 μM rosiglitazone, all obtained from Sigma-Aldrich. Adipocyte differentiation was initiated after a two-day culture period with insulin, followed by an additional eight days of culture with regular medium changes every two days. To evaluate potential anti-adipogenic effects, *Citrullus mucosospermus* extract (CME) at concentrations of 1, 10, or 25 μg/mL was introduced to the cells two hours prior to exposure to the DMI differentiation medium. The extract was continuously present throughout the eight-day differentiation period.

### 2.5. Cell Viability

To assess cell viability, adipocytes were seeded overnight in 96-well plates at a density of 1 × 10^4^ cells per well. Once the cells adhered, they were treated with various concentrations of CME for 24 h. Afterward, CCK-8 solution was added and the cells were incubated for 4 h. Absorbance at 450 nm was then measured using a microplate reader to determine cell viability.

### 2.6. Oil Red O Staining

3T3-L1 cells were cultured in Dulbecco’s Modified Eagle’s Medium (DMEM) supplemented with 10% fetal bovine serum (FBS). The cells were maintained at 37 °C in a humidified atmosphere with 5% CO_2_ until they reached approximately 80% confluence. Prior to experimentation, the cells were washed three times with cold phosphate-buffered saline (PBS) to remove any residual media. The washed cells were fixed by incubating them with 4% paraformaldehyde for 30 min at room temperature. This step was crucial to preserve the cellular morphology for subsequent analysis. Following fixation, the cells were stained with 0.5% Oil Red O solution for 30 min. Oil Red O binds to intracellular lipid droplets, enabling visualization and the quantification of lipid accumulation. After staining, the excess Oil Red O stain was removed by washing the cells three times with distilled water. The stained cells were then examined under an optical microscope at 200× magnification (Motic, Xiamen, China). Images were captured to document Oil Red *O*-positive lipid droplets within the cells. To quantify the amount of accumulated lipids, Oil Red O-positive cells were treated with isopropanol to extract the dye. The absorbance of the extracted solution was measured at 520 nm using a Wallac VICTOR plate reader (Perkin Elmer Corp., Waltham, MA, USA).

### 2.7. Glucose Uptake Measurement

3T3-L1 cells were seeded and differentiated in 24-well plates as described in Section 2.4. The differentiation process involved treatment with a differentiation medium (DMI) containing insulin, dexamethasone, and rosiglitazone over an eight-day period. On day 8 of differentiation, the culture supernatant from each well was carefully collected. The collected supernatants were centrifuged at 1000× *g* for 5 min to remove any cellular debris. The concentration of glucose in the clarified supernatants was determined using a commercial glucose detection kit (Asan Pharmaceutical, Seoul, Republic of Korea). The quantification was performed according to the manufacturer’s instructions, ensuring an accurate measurement of glucose levels in the cell culture medium.

### 2.8. Triglyceride (TG) Measurement

After inducing differentiation of 3T3-L1 cells in 24-well plates, as outlined in Section 2.4, triglycerides (TGs) were extracted on day 8 using 5% Triton X-100 (Bioshop, Burlington, ON, Canada). On day 8 post-differentiation, the culture medium was aspirated from each well. Cells were then washed once with phosphate-buffered saline (PBS). Subsequently, 100–200 μL of 5% Triton X-100 solution was added to each well to lyse the cells and solubilize intracellular lipids, including triglycerides. The plate was gently agitated or swirled to ensure a thorough mixing of the Triton X-100 solution with cellular contents. The plate was then incubated at room temperature for a suitable period (typically 10–15 min) to allow complete cell lysis and lipid extraction. Following incubation, the lysate containing extracted triglycerides was transferred to microcentrifuge tubes. Triglyceride levels were quantified using a commercial triglyceride detection kit (Asan Pharmaceutical, Seoul, Republic of Korea), following the manufacturer’s protocol.

### 2.9. Animal Anti-Obesity Study

The animal study investigating anti-obesity effects adhered to rigorous ethical standards approved by the Pusan National University Institutional Animal Care and Use Committee (PNU-IACUC approval code: PNU-2022-0139, approval date: 17 March 2022). Forty-seven 7-week-old male C57BL/6 mice were procured from Samtako Bio-Korea, Inc. (Osan, Republic of Korea). They were housed at the Pusan National University Laboratory Animal Resources Center, accredited by KFDA and AAALAC International. The mice were maintained under specific pathogen-free conditions with a 12 h light/dark cycle, a constant temperature of 23 ± 2 °C, and a relative humidity of 50 ± 10%. Animals had ad libitum access to standard chow and water throughout the study period. Mice were divided into two main groups: Non-high-fat diet (non-HFD, *n* = 10) who received a 10% kcal fat diet and high-fat diet (HFD, *n* = 37) who were fed a 60% kcal fat diet sourced from Research Diets. Within the HFD group, mice were further subdivided into the HFD+Po group, who were given Orlistat (10 mg/kg body weight daily) via oral administration and the HFD+Ve group, who were given distilled water control via oral administration. The HFD+CME group (1, 10, and 25 mg/kg body weight daily) were given *Citrullus mucosospermus* extract at varying doses via oral administration (Figure 1). The body weights of the mice were measured twice weekly throughout the study. Weekly assessments of food and water intake were conducted according to KFDA guidelines. At the end of the study period, mice were euthanized using CO_2_. Tissue samples, including organs of interest, were collected, preserved in Eppendorf tubes, and stored at −70 °C for subsequent biochemical and molecular analyses.

### 2.10. Histopathological Analysis

Liver and adipose tissue samples obtained from mice underwent a standardized preparation process for a histological analysis. The samples were initially immersed in 10% formalin for 48 h to preserve tissue architecture. Following fixation, the tissues were dehydrated using ethanol and embedded in paraffin wax. This process ensures the tissues are preserved and can be thinly sliced for microscopy. The paraffin-embedded tissues were cut into slices of 4 μm thickness using a microtome. These thin sections facilitate detailed microscopic examination. The tissue sections were stained with hematoxylin and eosin (H and E) using reagents sourced from Sigma-Aldrich. Hematoxylin stains nuclei blue-purple, while eosin stains cytoplasm and extracellular matrix pink, enabling cellular and tissue structure visualization. Stained tissue sections were examined using optical microscopy provided by Leica Microsystems (Wetzlar, Germany).

### 2.11. Real-Time Quantitative PCR Analysis

TNF-α, IL-1β, IL-6, and NF-κB mRNA expression in liver tissue was measured by real-time quantitative PCR. Mid-colon tissues were homogenized in RNA Bee solution (Tet-Test Inc., Friendswood, TX, USA) to extract total RNA. RNA Bee solution is effective in disrupting cells and solubilizing RNA. The extracted RNA samples were then purified to remove contaminants, and were eluted in RNase-free water. The concentration and purity of the extracted RNA were determined spectrophotometrically, typically using a UV–Vis spectrophotometer. This step ensures that RNA of sufficient quality and quantity is obtained for downstream applications. Complementary DNA (cDNA) was synthesized from the extracted RNA using a reverse transcription reaction. Deoxyribonucleotide triphosphates (dNTPs) were used as nucleotide building blocks. Superscript II reverse transcriptase (Thermo Fisher Scientific Inc., Waltham, MA, USA), which catalyzes the synthesis of cDNA from RNA templates, was used. Oligo-dT primers (Thermo Fisher Scientific Inc.) that anneal to the poly-A tail of mRNA were used, initiating cDNA synthesis. The synthesized cDNA served as the template for qPCR, which allows for the quantification of specific RNA sequences. qPCR reactions were prepared using 2× Power SYBR Green PCR Master Mix (Toyobo Co., Ltd., Osaka, Japan), which includes SYBR Green dye for the fluorescent detection of PCR products.

### 2.12. Statistical Analysis

The results were presented as the mean ± standard deviation obtained from a minimum of three independent experimental replicates. A two-way analysis of variance (ANOVA) was employed to assess variances between groups, followed by Duncan’s multiple range post hoc test using SPSS 20.0 software. A *p*-value less than 0.05 was considered statistically significant. This rigorous approach allowed for a robust evaluation of the experimental data and reliable interpretation of group differences.

## 3. Results

### 3.1. Inhibition of Adipogenesis by CME in 3T3-L1 Preadipocytes

The impact of CME on the viability of 3T3-L1 preadipocytes was assessed using the CCK-8 assay. Within the concentration range of 10–200 µg/mL, CME did not exhibit any cytotoxic effects. Furthermore, cell viability remained stable across this concentration range, with no significant concentration-dependent variations observed (Figure 2A). We investigated how CME modulates the differentiation of 3T3-L1 preadipocytes into mature adipocytes. Typically, when induced to differentiate, 3T3-L1 cells undergo lipid accumulation and an increase in cell size. Upon treatment with the differentiation-inducing DMI, control cells exhibited numerous small lipid droplets characteristic of mature adipocytes. However, when CME was co-administered along with DMI, there was a remarkable reduction in lipid droplet formation. Notably, this inhibitory effect of CME on adipogenesis displayed a dose-dependent pattern. Higher concentrations of CME led to a more potent suppression of lipid droplet accumulation in the cells. Microscopic examination at 200× magnification clearly visualized the differences. The DMI control group showed cells filled with numerous tiny lipid droplets resembling crimson balloons. In contrast, cells treated with CME had far fewer lipid droplets, with the highest CME dose virtually abolishing their formation. A quantitative analysis by measuring Oil Red O staining intensity at 520 nm further confirmed the dose-dependent decrease in lipid content upon CME treatment compared to the DMI control (Figure 2B).

### 3.2. CME Mitigates DMI-Induced Metabolic Alterations in 3T3-L1 Adipocytes

As expected, DMI treatment alone led to a substantial increase in glucose uptake and intracellular triglyceride (TG) accumulation, indicative of mature adipocyte metabolism. Remarkably, co-treatment with CME resulted in a dose-dependent attenuation of these metabolic alterations (Figure 3). Higher CME doses more potently reduced both TG accumulation and glucose uptake in the DMI-differentiated adipocytes. This ability of CME to modulate metabolic pathways in mature adipocytes suggests its potential for targeting obesity-associated metabolic dysregulation.

### 3.3. Effect of CME on Body Weight Regulation in HFD-Fed C57BL/6N Mice

To explore the potential anti-obesity effects of CME, C57BL/6N mice were subjected to a 10-week high-fat diet (HFD), known to induce weight gain. The weight trajectories during this period revealed distinct patterns: mice on HFD alone exhibited consistent weight gain, while those receiving concurrent CME treatment alongside HFD showed a significant reduction in weight gain, indicating potential anti-obesity effects (Table 1). The liver, crucial for metabolism and significantly affected by HFD, exhibited a notable decrease in weight following CME treatment compared to the HFD control group (Table 1). In contrast, there were no significant differences in kidney weight between the CME-treated group and the HFD control group (Table 1). Furthermore, CME administration led to a significant decrease in total fat weight compared to the HFD control group (Table 1). Specifically, visceral fat, located deep within the abdomen and associated with health risks, showed a marked reduction in weight with CME treatment (Table 1). Similarly, retroperitoneal fat, another critical site for fat storage, also displayed a substantial decrease in mice treated with CME compared to the HFD control group (Table 1).

### 3.4. Effect of CME on Hepatic Fat Accumulation in C57BL/6N Mice

To investigate the impact of CME on hepatic fat accumulation, C57BL/6N mice were assigned to various dietary groups. The control group received a standard diet, while the experimental groups were fed a high-fat diet (HFD) known to induce fatty liver disease. Additional HFD groups were supplemented with 1 (CME1), 10 (CME10), and 25 (CME25) mg/kg of CME to assess its mitigating effects. Orlistat, a recognized anti-obesity drug, was served as a positive control. A histological analysis of hepatic sections stained with hematoxylin and eosin revealed substantial steatosis in the HFD group, characterized by large lipid droplets occupying significant liver parenchyma. While Orlistat exhibited moderate efficacy in reducing lipid droplet number and size, its effect was less pronounced compared to CME. Notably, CME administration dose-dependently reduced hepatic fat accumulation. CME1 demonstrated noticeable reductions in both the number and size of lipid droplets compared to the HFD control group. CME10 further enhanced this effect, resulting in a liver architecture with fewer and smaller lipid droplets. The highest dose, CME25, produced the most significant effect, nearly eliminating steatosis and restoring liver morphology comparable to the control group (Figure 4).

### 3.5. Evaluation of CME’s Impact on Abdominal White Adipose Tissue

In-depth scrutiny of the effects of CME on adipose tissue led us to focus on the abdominal white adipose tissue as a central depot for lipid storage. Mice subjected to distinct dietary conditions, including control, HFD, HFD supplemented with Orlistat (an approved anti-obesity drug), and HFD with various doses of CME, underwent a microscopic analysis to assess the efficacy against hypertrophied adipocytes. Using hematoxylin and eosin staining as a microscopic gauge, we meticulously quantified the adipocytes. In the HFD group, these adipocytes asserted dominance and sprawled across the landscape. Although Orlistat demonstrated some efficacy in restraint, it failed to completely deflate inflated adipocytes. At the lowest dose (CME1), a discernible reduction in adipocyte size was evident, indicating the gradual regression of their distended morphology. CME10, a moderate contender, imparted a more potent effect, further reducing the adipocyte dimensions. Finally, the high-dose champion, CME25, emerged victorious, inducing a significant reduction in adipocyte size compared to the HFD group, creating an environment characterized by smaller and more tranquil adversaries (Figure 5).

### 3.6. Impact of CME on HFD-Induced Hepatic Inflammation

The development of fatty liver disease is closely associated with inflammation, where specific cytokines play critical roles in driving damaging processes. TNF-α, IL-1β, IL-6, and NF-κB are key players in orchestrating inflammatory responses that can lead to liver damage, fibrosis, and cirrhosis. Understanding their expression patterns during exposure to a high-fat diet (HFD) is crucial for developing effective strategies against this harmful progression. In our study, we investigated the impact of CME on HFD-induced hepatic inflammation in C57BL/6N mice. Mice fed an HFD showed significant increases in hepatic mRNA levels of TNF-α, IL-1β, IL-6, and NF-κB compared to control mice. Importantly, treatment with CME exerted a substantial influence by reducing the mRNA levels of all four cytokines in HFD-fed mice (Figure 6). This pivotal finding highlights the potential of CME in alleviating the inflammatory environment associated with fatty liver disease.

## 4. Discussion

*Citrullus mucosospermus*, a plant belonging to the Cucurbitaceae family, has garnered interest for its potential health benefits due to the presence of various bioactive compounds. These compounds act as nature’s tiny powerhouses, offering a range of effects on the body. Among these, cucurbitacins, particularly cucurbitacin E and its variations, are noteworthy. Cucurbitacins are widespread in the Cucurbitaceae family and possess diverse properties, potentially protecting the liver, reducing inflammation, fighting cancer, and regulating blood sugar. Cucurbitacin E is especially notable for its strong antioxidant and anti-inflammatory abilities, including the inhibition of inflammatory molecules like TNF-alpha and IL-6 [20]. Another bioactive compound found in *C. mucosospermus* is cucurbitacin E-2-O-glucoside. Although less studied than its aglycone form, this compound shows promising therapeutic potential. Further research on cucurbitacin glycosides from Citrullus species might unveil new bioactive compounds and shed light on how their structure influences their effects. In addition to cucurbitacins, *C. mucosospermus* is rich in phenolic acids, flavonoids, and other terpenoids. These additional bioactive components have been linked to suppressing fat cell formation, aiding weight management, and improving cholesterol levels in obesity models. Phenolic compounds like chlorogenic acid and flavonoid glycosides can hinder the development of pre-fat cells, while terpenoids like lycopene and cucurbitane triterpenoids can enhance fatty acid oxidation and inhibit lipogenesis [21,22]. The potential health benefits of *C. mucosospermus* may arise from the combined effects of these various bioactive compounds. Together, they may provide antioxidant, anti-inflammatory, and fat metabolism regulation properties, making this plant a promising candidate for obesity treatment [23,24,25]. Our exploration of the therapeutic potential of CME in countering adipogenesis and associated metabolic alterations in murine models has yielded multifaceted insights, significantly contributing to our understanding of anti-obesity strategies. This study aimed to elucidate the effects of CME on adipogenesis, adipocyte metabolism, and inflammation in both in vitro and in vivo models, revealing promising results that warrant further discussion.

The use of the 3T3-L1 preadipocyte model allowed for an in-depth examination of the impact of CME on early adipocyte differentiation. The observed dose-dependent reduction in lipid droplet accumulation underscores the potent anti-adipogenic properties of CME. This inhibitory effect on adipogenesis has critical implications, as dysregulated adipocyte differentiation is a hallmark of obesity [28,29]. Beyond anti-adipogenesis, CME showed a remarkable ability to modulate metabolic alterations in fully differentiated adipocytes. The dose-dependent attenuation of glucose uptake and intracellular triglyceride accumulation suggests a regulatory role in adipocyte metabolism. This intricate modulation of metabolic pathways positions CME as a promising candidate for interventions targeting obesity-associated metabolic dysregulation. Translating our in vitro findings into an in vivo context using C57BL/6N mice fed an HFD revealed the compelling anti-obesity effects of CME. Concurrent administration of CME with an HFD resulted in significant reductions in body weight gain, liver weight, and total fat weight. The specific decrease in visceral and retroperitoneal fat deposits emphasizes the targeted efficacy of CME in addressing adipose tissue expansion, a hallmark of obesity. Histological analyses of liver tissues revealed a significant impact of CME on hepatic fat accumulation, surpassing the efficacy of the established anti-obesity drug Orlistat. The dose-dependent reduction in hepatic lipid droplet numbers and sizes highlights the anti-steatotic potential of CME and its potential for addressing fatty liver disease associated with obesity. Chronic inflammation is a key contributor to the progression of obesity-related complications [30,31]. CME exhibited a significant modulatory effect on hepatic inflammatory responses in HFD-fed mice. The reduction in the mRNA levels of pro-inflammatory cytokines (TNF-α, IL-6, and IL-1β) signifies a robust anti-inflammatory effect of CME, offering a potential mechanism for its protective role against hepatic inflammation associated with obesity.

While our study provides substantial evidence supporting the therapeutic potential of CME, it is essential to acknowledge certain limitations. Notably, the study design does not allow for the exclusion of the possibility that the observed effects on liver, fat, and inflammatory markers are secondary to weight loss rather than direct effects of CME. Therefore, it is crucial to consider that the beneficial outcomes observed might be partially or wholly attributed to the weight reduction induced by CME rather than direct pharmacological actions on these markers. Additionally, the current study primarily focuses on preclinical models, and the translatability of these findings to human subjects remains to be established. Further clinical studies are needed to confirm the efficacy and safety of CME in human populations. Future research endeavors should delve deeper into unraveling the precise molecular mechanisms underlying CME’s therapeutic actions. This includes exploring the specific signaling pathways modulated by CME and identifying the bioactive compounds responsible for its anti-obesity effects. Moreover, rigorous evaluations of its safety profile through long-term toxicity studies and further explorations of its efficacy in clinical settings will be pivotal in advancing CME towards potential clinical applications for treating obesity and its associated metabolic complications.

## 5. Conclusions

This study establishes *Citrullus mucosospermus* extract (CME) as a promising candidate in the realm of obesity therapy and the management of associated metabolic complications. Through comprehensive investigations using in vitro and in vivo models, CME demonstrated robust efficacy as an anti-adipogenic, anti-obesity, and anti-steatotic agent. Importantly, our findings highlight CME’s ability to modulate inflammatory pathways and metabolic processes, suggesting its potential as a multifaceted therapeutic approach against obesity-related disorders. The observed dose-dependent reduction in adipogenesis and adiposity in 3T3-L1 adipocytes and HFD-fed C57BL/6N mice underscores the pharmacological potency of CME. Furthermore, CME’s ability to attenuate hepatic steatosis and reduce pro-inflammatory cytokine expression in the liver points towards its beneficial effects on metabolic health. Moving forward, future research endeavors should delve deeper into unraveling the precise molecular mechanisms underlying CME’s therapeutic actions. Additionally, rigorous evaluations of its safety profile and further explorations of its efficacy in clinical settings will be pivotal in advancing CME towards potential clinical applications for treating obesity and its associated metabolic complications.

## Figures and Tables

**Figure 1 nutrients-16-02171-f001:**
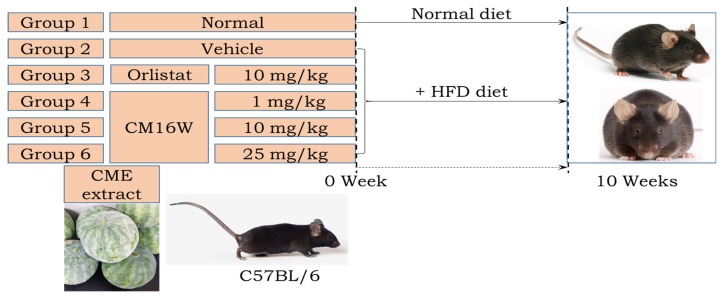
Experimental design for investigating the effects of *Citrullus mucosospermus* extract on an HFD diet.

**Figure 2 nutrients-16-02171-f002:**
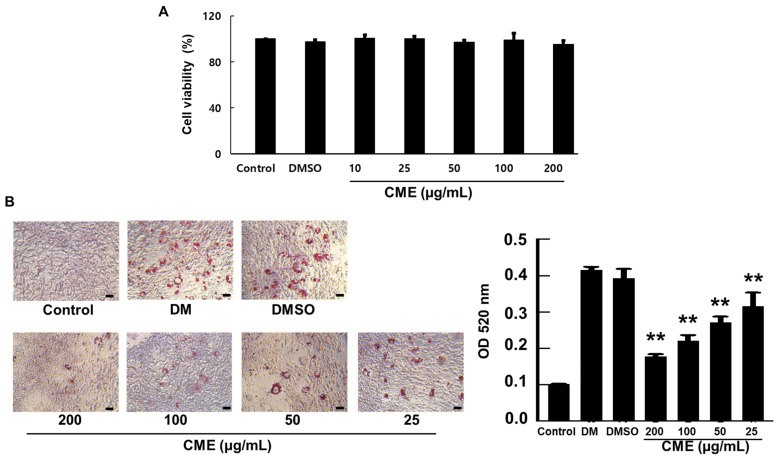
CME suppresses lipid droplet accumulation and adipogenesis in 3T3-L1 cells. 3T3-L1 preadipocytes were treated with differentiation medium (DMI) alone or DMI supplemented with varying concentrations of CME for 8 days. (**A**) Demonstrates the effects of CME on the vitality of adipocytes. (**B**) Representative microscopic images (200×) on day 8 showing a dose-dependent decrease in lipid droplet formation in CME-treated cells compared to DMI control, as visualized by Oil Red O staining (scale bar = 20 μm). A quantitative analysis of lipid content by measuring absorbance of extracted Oil Red O at 520 nm confirmed the reduction with increasing CME concentrations. Cells treated with 0.1% dimethyl sulfoxide were used as negative controls. Data are mean ± SD from three independent experiments. ** *p* < 0.01 versus DMI control.

**Figure 3 nutrients-16-02171-f003:**
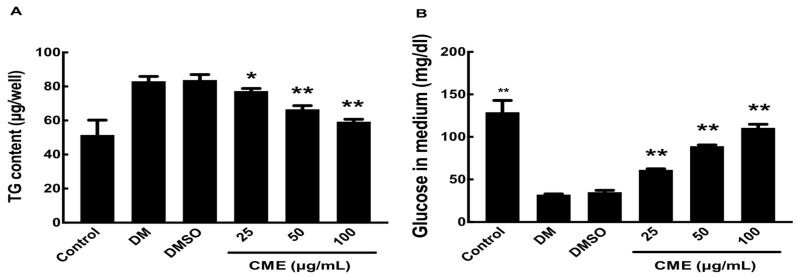
Impact of CME on glucose uptake and triglyceride storage in differentiated 3T3-L1 adipocytes. 3T3-L1 cells were induced to differentiate into adipocytes using DMI, either alone or in combination with different concentrations of CME over an 8-day period. (**A**) On the 8th day, culture supernatants were harvested and glucose uptake was assessed with a commercial assay kit. (**B**) Intracellular triglycerides were measured by extracting lipids with Triton X-100, followed by an analysis with a triglyceride assay kit. Cells treated with 0.1% dimethyl sulfoxide were used as negative controls. Data are presented as mean ± SD from three separate experiments, with statistical significance denoted as * *p* < 0.05 and ** *p* < 0.01 versus DMI control.

**Figure 4 nutrients-16-02171-f004:**
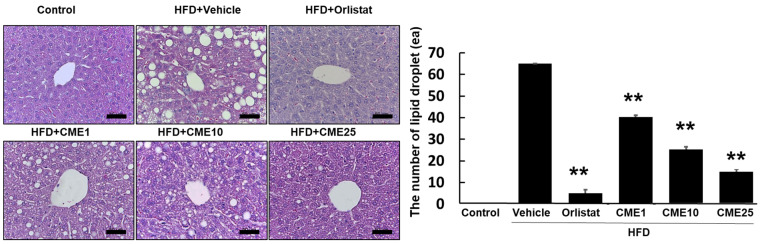
Effect of CME on hepatic steatosis in HFD-fed C57BL/6N mice. Liver sections stained with hematoxylin and eosin were examined to assess hepatic steatosis in C57BL/6N mice fed different diets: control diet, HFD alone (Vehicle), HFD + Orlistat (Orlistat), and HFD + varying doses of CME (CME1, CME10, and CME25; 1 mg/kg, 10 mg/kg, and 20 mg/kg body weight CME daily via oral administration). scale bar = 50 μm. Each panel represents observations from *n* = 10 mice per group. ** *p* < 0.01 versus HFD group.

**Figure 5 nutrients-16-02171-f005:**
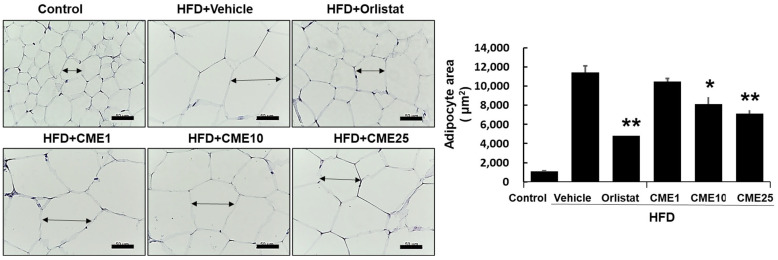
Effect of CME on adipocyte hypertrophy in the abdominal white adipose tissue of HFD-fed mice. Sections of abdominal white adipose tissue were stained with hematoxylin and eosin to assess adipocyte hypertrophy in mice fed different diets: control diet, HFD alone (Vehicle), HFD + Orlistat (Orlistat), and HFD + varying doses of CME (CME1, CME10, and CME25; 1 mg/kg, 10 mg/kg, and 20 mg/kg body weight CME daily via oral administration). A microscopic examination at 100× magnification revealed hypertrophied adipocytes in the HFD group. Each panel represents observations from *n* = 10 mice per group. * *p* < 0.05 and ** *p* < 0.01 versus HFD group.

**Figure 6 nutrients-16-02171-f006:**
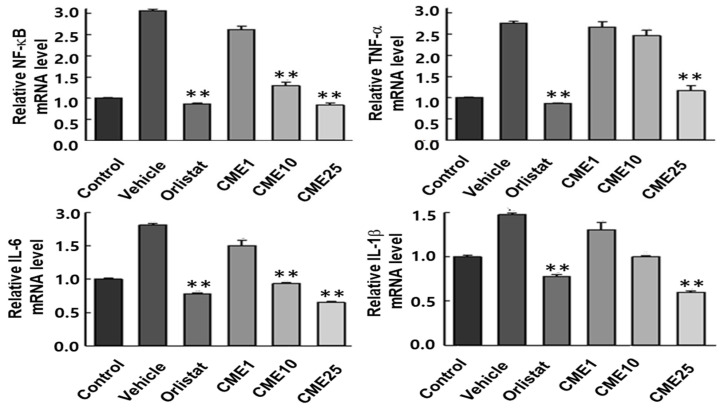
Effect of CME on pro-inflammatory cytokine expression in the livers of HFD-fed C57BL/6 mice. Hepatic mRNA levels of TNF-α, IL-6, IL-1β, and NF-κB were assessed by real-time qPCR in liver tissues from mice fed a control diet or HFD, with or without CME supplementation. Data are presented as mean ± SEM, *n* = 10 per group. ** *p* < 0.01 versus HFD group.

**Table 1 nutrients-16-02171-t001:** Effects of CME on body weight regulation and adipose tissue expansion in HFD-fed C57BL/6N mice.

Group	Control	HFD	Orlistat	CME1	CME10	CME25
Initial body weight (g)	17.02 ± 0.39	17.05 ± 0.23	17.13 ± 0.32	17.03 ± 0.20	16.94 ± 0.21	17.07 ± 0.35
Final body weight (g)	26.9 ± 0.9	38.9 ± 1.7	30.7 ± 1.3 **	36.8 ± 1.4 *	34.7 ± 1.1 **	32.8 ± 1.7 **
Food intake (g·day^−1^)	2.25 ± 0.24	2.29 ± 0.13	1.83 ± 0.10 **	2.02 ± 0.11 *	1.95 ± 0.23 *	1.85 ± 0.37 **
Liver weight (g·mouse^−1^)	0.96 ± 0.03	1.39 ± 0.10	1.16 ± 0.06	1.33 ± 0.09	1.19 ± 0.04	1.21 ± 0.08
Kidney weight (g·mouse^−1^)	0.34 ± 0.01	0.39 ± 0.01	0.40 ± 0.0	0.40 ± 0.01	0.39 ± 0.01	0.40 ± 0.01
Fat weight (g·mouse^−1^)	0.82 ± 0.12	3.12 ± 0.21	2.13 ± 0.31 **	2.78 ± 0.22 *	2.47 ± 0.25 *	2.37 ± 0.22 *
Abdominal fat weight (g·mouse^−1^)	0.54 ± 0.10	2.27 ± 0.15	1.49 ± 0.23 **	2.01 ± 0.16 *	1.88 ± 0.22 *	1.75 ± 0.18 **
Retroperitoneal fat weight (g·mouse^−1^)	0.22 ± 0.06	0.85 ± 0.07	0.64 ± 0.09 **	0.77 ± 0.07 *	0.60 ± 0.07 *	0.62 ± 0.08 *

Body weight changes over a 10-week period in mice fed a normal diet (control), high-fat diet (HFD, Vehicle), or HFD supplemented with varying doses of CME (1 mg/kg, 10 mg/kg, and 20 mg/kg body weight CME daily via oral administration) or the anti-obesity drug Orlistat (positive control). Liver weight measurements following 10 weeks of dietary intervention are shown. Kidney weight comparisons among experimental groups are shown. The total fat weight analysis at the end of the study period is shown. An assessment of abdominal fat deposits in different treatment groups is shown. An evaluation of retroperitoneal fat accumulation in mice from each experimental group is shown. Data are presented as mean ± SD, with *n* = 10 per group. Statistical significance denoted as * *p* < 0.05 and ** *p* < 0.01 versus the HFD group.

## Data Availability

The data presented in this study are available on request from the corresponding author due to privacy, legal, or ethical restrictions.

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
