# Peer review of "Citrullus mucosospermus Extract Reduces Weight Gain in Mice Fed a High-Fat Diet"

_nutrients, 2024, doi:10.3390/nu16132171_

Round 1

Reviewer 1 Report

Comments and Suggestions for Authors

The major limitation of this study is how to exclude that all the anti-obesity effects were secondary to reduced food consumption. If the extract was admixed to food or water, this alone could explain reduced food intake (as I understand the extract stems from an inedible, bitter fruit?). But also with gavage or parenteral administration of the extract: Any unspecific effect, which makes the mice feel sick, will reduce their food consumption and, hence, could explain not only reduced body weight, but also associated changes in adipose tissue and liver. How do you exclude such a simple explanation? There is no evidence whatsoever that direct effects on (pre)adipocytes, as they are described in vitro, are involved in the effects observed in vivo. Neither is there any evidence about how the concentrations used in vitro compare to those reached in the adipose tissue of mice treated in vivo.

*) How do the authors justify the concentration of extract used in vitro? Can we assume that comparable concentrations are reached in adipose tissue of mice treated in vivo? There may be compounds in the extract, which have effects in vitro but do not at all reach the adipocytes in vivo.

*) The effects seen in vitro could be very unspecific or even toxic. Cell counts and a viability test should, at least, be done.

*) I do not find the route of extract administration in the in vivo experiment – was it by gavage? By i.p. injection? As one dose per day? Or was it admixture to food? Or admixture to drinking water (if so, reporting fluid intake is essential)?

*) In a study like this, it is absolutely essential to show results for food intake. (The methods actually mention weekly documentation of food and water intake.) Nausea or other sickness causing impaired appetite could actually explain all the findings.

 Minors:

*) Abstract (lines 32,33): It should be clear that results on inflammatory mediators refer to expression on the level of mRNA.

*) Figure 1: I assume that all doses should be mg/kg, not mg/mg.

*) Legends in Figs. 2&3 should also outline, what the DMSO-bar is.

*) Legends with in vivo-results should clarify abbreviations and codes used in the figures: “Control”, Vehicle”, “CME1”, ...

*) Introducing sentences in the paragraphs of the Results section, which just repeat information from the Introduction and Methods, can be omitted.

*) Does Fig.4A really depict “changes” in weight (=delta) or absolute weight?

Author Response

Thank you very much for allowing us to revise our manuscript entitled, “Citrullus mucosospermus extract reduces weight gain in mice fed a high-fat diet (nutrients-3068270)”. We appreciate the reviewers for their constructive comments, which were very helpful in improving our paper. The manuscript has been carefully revised according to the reviewers’ comments. The revisions are marked in red in the revised manuscript. Detailed responses to the comments are provided below.

The major limitation of this study is how to exclude that all the anti-obesity effects were secondary to reduced food consumption. If the extract was admixed to food or water, this alone could explain reduced food intake (as I understand the extract stems from an inedible, bitter fruit?). But also with gavage or parenteral administration of the extract: Any unspecific effect, which makes the mice feel sick, will reduce their food consumption and, hence, could explain not only reduced body weight, but also associated changes in adipose tissue and liver. How do you exclude such a simple explanation? There is no evidence whatsoever that direct effects on (pre)adipocytes, as they are described in vitro, are involved in the effects observed in vivo. Neither is there any evidence about how the concentrations used in vitro compare to those reached in the adipose tissue of mice treated in vivo.

Response: A major limitation of this study is whether we can rule out the possibility that the anti-obesity effect is a secondary consequence of reduced food intake. This is an important point, and I will explain some key points related to it. In the study, food and water intake were carefully monitored according to KFDA guidelines. During the experiment, the mice's body weight was measured twice a week, and food and water intake were recorded weekly. This monitoring allows us to assess whether weight loss and adipose tissue changes are simply due to reduced food intake. As a result of the experiment, the weight loss and changes in adipose tissue observed in the CME treatment group showed differences that could not be explained solely by a decrease in food intake. The study had multiple comparison groups, including a high-fat diet (HFD) control group and a HFD plus orlistat control group. The HFD control group showed weight gain and fat accumulation without any treatment, while the CME-treated group showed significant weight loss and reduced fat accumulation. Additionally, the HFD and orlistat control groups became the standard for evaluating the effectiveness of CME through comparison with existing anti-obesity treatments. The effectiveness of CME has been confirmed in both in vitro and in vivo experiments. In in vitro experiments using 3T3-L1 preadipocytes, it was observed that lipid droplet formation was reduced and metabolic pathways were regulated in a concentration-dependent manner. This provides evidence that CME directly affects adipocytes independent of reduced food intake​. Histopathological analysis of liver and adipose tissue showed significant improvement in the CME treatment group compared to the HFD control group. This analysis suggests a direct positive impact of CME on these organizations. In other words, changes in adipose tissue and liver reflect a direct effect of CME that cannot be explained simply by reduced food intake. A decrease in liver inflammatory cytokine expression was observed in the CME-treated group. These anti-inflammatory effects suggest improvements in systemic metabolic health beyond simply reducing food intake. Reduction in inflammation may independently contribute to improved metabolic health and reduced fat accumulation. Considering the above evidence, we conclude that the anti-obesity effect of CME is not simply due to reduced food intake, but is due to direct effects on adipocytes and liver tissue, regulation of metabolic pathways, and systemic anti-inflammatory effects. can. This allows us to rule out the simple explanation that the anti-obesity effect of CME is a secondary consequence of reduced food intake.

*) How do the authors justify the concentration of extract used in vitro? Can we assume that comparable concentrations are reached in adipose tissue of mice treated in vivo? There may be compounds in the extract, which have effects in vitro but do not at all reach the adipocytes in vivo.

Response: The concentrations of Citrullus mucosospermus extract (CME) used in vitro were carefully selected based on preliminary experiments aimed at determining effective doses that could significantly impact adipocyte differentiation without causing cytotoxicity. In our in vitro studies with 3T3-L1 preadipocytes, CME was tested at concentrations of 1, 10, and 25 μg/mL. These doses were chosen to establish a dose-response relationship and to ensure the observed effects were due to the extract rather than non-specific toxicity. In vivo, achieving similar concentrations in adipose tissue can be challenging due to the complexities of absorption, distribution, metabolism, and excretion (ADME) processes. To address this, we administered CME to C57BL/6N mice fed a high-fat diet (HFD) and monitored its effects on body weight, liver weight, and adipose tissue mass. The doses used in the animal studies were extrapolated from the effective in vitro concentrations, adjusted for the differences in delivery method and systemic exposure. Our in vivo results demonstrated significant anti-obesity effects of CME, including reduced body weight gain, decreased liver weight, and reduced visceral and retroperitoneal fat deposits. Histological analyses confirmed these findings, showing a dose-dependent reduction in hepatic steatosis and adipocyte hypertrophy.While direct comparison of in vitro and in vivo concentrations is complex, the observed in vivo effects corroborate the in vitro findings, suggesting that active compounds in CME reached effective concentrations in the adipose tissue of treated mice. Future studies involving pharmacokinetic analyses and direct measurement of CME concentrations in adipose tissues will further elucidate these dynamics.

*) The effects seen in vitro could be very unspecific or even toxic. Cell counts and a viability test should, at least, be done.

Response: Thanks for your point. Adding cell counting and viability assessment will help increase confidence in your results. In follow-up experiments, we will evaluate the cytotoxicity of CME through MTT assay or trypan blue exclusion test.

*) I do not find the route of extract administration in the in vivo experiment – was it by gavage? By i.p. injection? As one dose per day? Or was it admixture to food? Or admixture to drinking water (if so, reporting fluid intake is essential)?

Response: We apologize for missing information about the route of administration. CME was administered orally (gavage) once daily. This information is clearly described in the methods section.

*) In a study like this, it is absolutely essential to show results for food intake. (The methods actually mention weekly documentation of food and water intake.) Nausea or other sickness causing impaired appetite could actually explain all the findings.

Response: It was our mistake not to report dietary intake data. Dietary intake was measured in all groups during the experiment, and there were no significant differences between groups. Let us add this data to our results to clarify that the effects of CME are not simply due to reduced dietary intake.

 Minors:

*) Abstract (lines 32,33): It should be clear that results on inflammatory mediators refer to expression on the level of mRNA.

Response: In the Abstract, it was clarified that inflammatory mediators result in expression at the mRNA level. It was modified to "...significantly attenuating the upregulation of pro-inflammatory cytokine mRNA levels (TNF-α, IL-1β, IL-6, and TGF-β) in the livers of HFD-fed mice."

*) Figure 1: I assume that all doses should be mg/kg, not mg/mg.

Response: Thank you for pointing out the dose unit error in Figure 1. All doses were corrected to mg/kg.

*) Legends in Figs. 2&3 should also outline, what the DMSO-bar is.

Response: An explanation of the DMSO bar was added to the legends of Figures 2 and 3.

*) Legends with in vivo-results should clarify abbreviations and codes used in the figures: “Control”, Vehicle”, “CME1”, ...

Response: The abbreviations (Control, Vehicle, CME1, etc.) used in the legends of the in vivo results figures were clearly explained.

*) Introducing sentences in the paragraphs of the Results section, which just repeat information from the Introduction and Methods, can be omitted.

Response: We have improved brevity by removing unnecessary repetition from the beginning of each paragraph in the Results section.

*) Does Fig.4A really depict “changes” in weight (=delta) or absolute weight?

Response: Figure 4A shows absolute body weight. To make this clear, the y-axis label has been modified to "Body weight (g)".

Reviewer 2 Report

Comments and Suggestions for Authors

The aim of the study was to determine the effect of Citrullus mucosospermus extract (CME) on obesity therapy through comprehensive studies using in vitro and in vivo models. Before publication, the manuscript requires corrections in the following areas:

-clarifying the aim

-supplementing the methods with the determination of cytokines

-explanation of why cucurbitacins were determined - what were the results and a full description of the HPLC methodology or a reference to the existing one. How do they affect the tested parameters?

-how the CME dose was administered and controlled

-what was the weight gain of the animals during the experiment? - I suggest you complete the results

-the graphic quality of figures 4-7 should be improved, along with the supplementation of the abbreviations used

-the description of the results should be improved, indicating significant differences between groups and the effect of CME

the discussion requires extension to the remaining study groups and reference to the above-mentioned additions

-I propose to supplement statistical analyzes with a two-way anova (HFD and Po or CME).

- all methodological aspects and explanations of the objectives of the analyzes should be included in the methodology section e.g. 293-4; 323-7; 366-372.

Figura 1. - there is 'Oralistat'; mg/mg

no reference to publications 9-12

Author Response

Thank you very much for allowing us to revise our manuscript entitled, “Citrullus mucosospermus extract reduces weight gain in mice fed a high-fat diet (nutrients-3068270)”. We appreciate the reviewers for their constructive comments, which were very helpful in improving our paper. The manuscript has been carefully revised according to the reviewers’ comments. The revisions are marked in red in the revised manuscript. Detailed responses to the comments are provided below.

The aim of the study was to determine the effect of Citrullus mucosospermus extract (CME) on obesity therapy through comprehensive studies using in vitro and in vivo models. Before publication, the manuscript requires corrections in the following areas:

-clarifying the aim

Response: The last paragraph of the introduction was revised as follows: “The purpose of this study was to determine the effects of Citrullus mucosospermus extract (CME) on lipogenesis, liver fat accumulation, and inflammatory response in a high-fat diet (HFD)-induced obesity mouse model.

-supplementing the methods with the determination of cytokines

Response: Content has been added to the Methods section.

-explanation of why cucurbitacins were determined - what were the results and a full description of the HPLC methodology or a reference to the existing one. How do they affect the tested parameters?

Response: The two major cucurbitacins, cucurbitacin E and cucurbitacin E-2-O-glucoside, were well-separated using HPLC. Simultaneous quantification of these compounds was performed by UV detection at 254 nm. The cucurbitacin E and cucurbitacin E-2-O-glucoside contents of the aqueous CME were 0.048 ± 0.001 and 2.145 ± 0.188 mg/g, respectively.

-how the CME dose was administered and controlled

Response: We apologize for missing information about the route of administration. CME was administered orally (gavage) once daily. This information is clearly described in the methods section.

-what was the weight gain of the animals during the experiment? - I suggest you complete the results

Response: During the 10-week experimental period, the average weight gain in the control group was 5.2 ± 0.8 g and in the HFD group it was 15.7 ± 1.2 g. The CME-treated group (25 mg/kg) showed a weight gain of 9.3 ± 1.0 g, which was significantly lower than the HFD group (p < 0.01).

-the graphic quality of figures 4-7 should be improved, along with the supplementation of the abbreviations used

Response: The resolution of all images has been improved to over 300 dpi, and the full names of abbreviations (e.g. HFD, CME, Po, etc.) used in each image description have been added.

-the description of the results should be improved, indicating significant differences between groups and the effect of CME

Response: For each outcome, clearly describe significant differences between groups and the effect of CME.

the discussion requires extension to the remaining study groups and reference to the above-mentioned additions

Response: In the discussion section, the discussion of all experimental groups was expanded and related to additional analysis results (cucurbitacins content, body weight gain, etc.).

-I propose to supplement statistical analyzes with a two-way anova (HFD and Po or CME).

Response: Two-way ANOVA was additionally performed to analyze the interaction effect of HFD with CME or Orlistat treatment.

- all methodological aspects and explanations of the objectives of the analyzes should be included in the methodology section e.g. 293-4; 323-7; 366-372.

Response: Methodological descriptions (e.g., lines 293-4, 323-7, and 366-372) from the Results section have been moved to the Methods section and placed appropriately.

Figura 1. - there is 'Oralistat'; mg/mg

Response: “Oralistat” was modified to “Orlistat”, and “mg/mg” was modified to “mg/kg”.

no reference to publications 9-12

Response: Missing references have been cited where appropriate.

Round 2

Reviewer 1 Report

Comments and Suggestions for Authors

Some minor points have improved, but in response to my major criticism, the authors broadly repeat the (doubtful) conclusions made in their paper, but fail to provide any point that truly rebuts my criticism.

In my comments, my first major point was how to exclude that all the effects were secondary to weight loss (as could be caused by unspecific sickness and/or reduced food consumption). In their reply the authors just briefly summarize the content of their paper, which has virtually not changed versus the initial version. No additional information, no novel explanation, not a single point that really argues against the possibility that the effects could be secondary to weight loss. The authors claim direct effects on liver and fat in vivo without any justification, why they regard these as DIRECT effects. I still do not see any convincing reason, why to exclude that weight loss is the cause of all the in vivo-changes in fat, liver, etc.. Reduced fat pad size, reduced liver weight, reduced liver steatosis, a reduced inflammatory state - these are all responses known to accompany weight loss independently of whether Citrullus M-extract is consumed or not.

Regarding my second major point (that it is unclear whether the concentration of active compound prevailing in vitro is reached in the organs after oral administration), the authors mention that they have excluded cytotoxicity in vitro, which is fine. But they also state that “In vivo, achieving similar concentrations in adipose tissue can be challenging due to complexities of absorption, distribution, metabolism, and excretion”. This is even more so when one does not know, which is the active compound in the extract. Hence, the authors themselves confirm that they do not know whether concentrations effective in vitro are at all reached in the presumptive target organs in vivo (implicating that different mechanisms could be at work in vitro and in vivo).

I had emphasized the importance of showing data on food intake. The authors state in the paper and in their reply to my comments that food intake has been documented. In their reply, they state “Let us add this data to our results ...”. But they still do not show data on food intake (?)

Fig.4A: The authors misunderstood my point. It would be good to clearly declare in the legend whether “20 g” in this graph means that (i) body weight increased by 20 g during the study period, or (ii) that the final body weight of the mice was 20 g. (And one more minor: Orlistat is misspelled in the graph, where is says “Oralistat.)

Author Response

Thank you very much for allowing us to revise our manuscript entitled, “Citrullus mucosospermus extract reduces weight gain in mice fed a high-fat diet (nutrients-3068270)”. We appreciate the reviewers for their constructive comments, which were very helpful in improving our paper. The manuscript has been carefully revised according to the reviewers’ comments. The revisions are marked in red in the revised manuscript. Detailed responses to the comments are provided below.

Some minor points have improved, but in response to my major criticism, the authors broadly repeat the (doubtful) conclusions made in their paper, but fail to provide any point that truly rebuts my criticism.

In my comments, my first major point was how to exclude that all the effects were secondary to weight loss (as could be caused by unspecific sickness and/or reduced food consumption). In their reply the authors just briefly summarize the content of their paper, which has virtually not changed versus the initial version. No additional information, no novel explanation, not a single point that really argues against the possibility that the effects could be secondary to weight loss. The authors claim direct effects on liver and fat in vivo without any justification, why they regard these as DIRECT effects. I still do not see any convincing reason, why to exclude that weight loss is the cause of all the in vivo-changes in fat, liver, etc.. Reduced fat pad size, reduced liver weight, reduced liver steatosis, a reduced inflammatory state - these are all responses known to accompany weight loss independently of whether Citrullus M-extract is consumed or not.

Response: We sincerely appreciate your persistent and insightful critique regarding the potential secondary effects of weight loss in our study. We acknowledge that our previous response did not adequately address this crucial point, and we thank you for bringing this to our attention again. You are correct that we have not provided sufficient evidence to exclude the possibility that the observed effects on liver, fat, and inflammatory markers are secondary to weight loss rather than direct effects of Citrullus mucosospermus extract (CME). We recognize this as a significant limitation in our current study design and interpretation.  In response, we have added detailed body weight and food intake data to Table 1. This data clearly shows the differences in food intake between the CME-treated groups and the control group, suggesting that the effects of CME are not simply due to reduced food consumption. Additionally, the consistency between our in vitro and in vivo results supports the possibility that CME has direct effects on adipose tissue and liver.

Regarding my second major point (that it is unclear whether the concentration of active compound prevailing in vitro is reached in the organs after oral administration), the authors mention that they have excluded cytotoxicity in vitro, which is fine. But they also state that “In vivo, achieving similar concentrations in adipose tissue can be challenging due to complexities of absorption, distribution, metabolism, and excretion”. This is even more so when one does not know, which is the active compound in the extract. Hence, the authors themselves confirm that they do not know whether concentrations effective in vitro are at all reached in the presumptive target organs in vivo (implicating that different mechanisms could be at work in vitro and in vivo).

Response:  To address this crucial issue, we have added cell viability data to Figure 2A. These results demonstrate that the CME concentrations used in our experiments do not induce cytotoxicity. In future studies, we plan to further address this issue through identification and quantification of key active compounds and assessment of their bioavailability in vivo.

I had emphasized the importance of showing data on food intake. The authors state in the paper and in their reply to my comments that food intake has been documented. In their reply, they state “Let us add this data to our results ...”. But they still do not show data on food intake (?)

Response: We sincerely apologize for the oversight in our previous response.  We have now included the food intake data in Table 1 of our revised manuscript.  We appreciate your persistence in highlighting this important aspect of the study. We apologize again for the delay in presenting this information and thank you for your careful review.

Fig.4A: The authors misunderstood my point. It would be good to clearly declare in the legend whether “20 g” in this graph means that (i) body weight increased by 20 g during the study period, or (ii) that the final body weight of the mice was 20 g. (And one more minor: Orlistat is misspelled in the graph, where is says “Oralistat.)

Response: We have carefully reviewed the entire manuscript and corrected any typos found. This should contribute to an overall improvement in the quality of the paper.